XA21-specific induction of stress-related genes following Xanthomonas infection of detached rice leaves

Thomas Nicholas C. 1 2
Schwessinger Benjamin 1 2 3
Liu Furong 1 2
Chen Huamin 1 4
Wei Tong 1 2
Nguyen Yen P. 1
Shaker Isaac W.F. 1
Ronald Pamela C. pcronald@ucdavis.edu 1 2
1 Department of Plant Pathology and the Genome Center, University of California , Davis , CA , United States
2 Joint BioEnergy Institute , Emeryville , CA , United States
3 Research School of Biology, Australian National University , Acton , Australia
4 State Key Laboratory for Biology of Plant Diseases and Insect Pests, Institute of Plant Protection, Chinese Academy of Agricultural Sciences , Bejing , China
McCormick Sheila
Electronic publication date: 2016 Sep 28
Publication date: 2016
Volume: 4
Electronic Location ID: e2446
Received 2016 Jul 14; Accepted 2016 Aug 14
Copyright: ©2016 Thomas et al.
Copyright year: 2016
Copyright holder: Thomas et al.
License: This is an open access article distributed under the terms of the Creative Commons Attribution License, which permits unrestricted use, distribution, reproduction and adaptation in any medium and for any purpose provided that it is properly attributed. For attribution, the original author(s), title, publication source (PeerJ) and either DOI or URL of the article must be cited.
License URL: https://creativecommons.org/licenses/by/4.0/

Keywords: XA21, Xanthomonas, EFR, Rice, RNAseq

Funding: Lawrence Berkeley National Laboratory DOE Joint BioEnergy Institute US Department of Energy, Office of Science, Office of Biological and Environmental Research DE-AC02-05CH11231 Human Frontiers Science Program LT000674/2012 Discovery Early Career Award DE150101897 NIH GM59962 This work was part of the DOE Joint BioEnergy Institute (http://www.jbei.org) supported by the US Department of Energy, Office of Science, Office of Biological and Environmental Research, through contract DE-AC02-05CH11231 between Lawrence Berkeley National Laboratory and the US Department of Energy. This work was also supported by the NIH (GM59962). BS was supported by a Human Frontiers Science Program long-term postdoctoral fellowship (LT000674/2012) and a Discovery Early Career Award (DE150101897). The funders had no role in study design, data collection and analysis, decision to publish, or preparation of the manuscript.

==============================
The rice XA21 receptor kinase confers robust resistance to the bacterial pathogen Xanthomonas oryzaepv. oryzae (Xoo). We developed a detached leaf infection assay to quickly and reliably measure activation of the XA21-mediated immune response using genetic markers. We used RNA sequencing of elf18 treated EFR:XA21:GFP plants to identify candidate genes that could serve as markers for XA21 activation. From this analysis, we identified eight genes that are up-regulated in both in elf18 treated EFR:XA21:GFP rice leaves and Xoo infected XA21 rice leaves. These results provide a rapid and reliable method to assess bacterial-rice interactions.

Introduction

Plant immunity is mediated, in part, by cell surface immune receptors that recognize molecules produced by microbes. For example, the Arabidopsis FLS2 (Flagellin Sensing 2) and EFR (Elongation Factor Tu Receptor) receptors recognize the flg22 peptide derived from bacterial flagellin and the elf18 peptide derived from elongation factor thermo-unstable (EF-Tu) protein, respectively (Gomez-Gomez & Boller, 2000; Zipfel et al., 2006). The rice XA21 receptor recognizes the sulfated RaxX peptide (RaxX21-sY) derived from the RaxX protein produced by Xanthomonas oryzae pv. oryzae (Xoo) (Song et al., 1995; Pruitt et al., 2015; Wei et al., 2016). XA21, EFR, and FLS2 all contain extracellular leucine rich repeat (LRR), transmembrane, and intracellular non-RD (arginine-aspartic acid) kinase domains. These receptor domains are partially interchangeable. For example, the LRR domain from EFR can be fused to the transmembrane and intracellular domain of FLS2 to form a chimeric receptor that responds to elf18 treatments when transiently expressed in Nicotiana benthamiana and Arabidopsis thaliana (Albert et al., 2010). The EFR LRR can be fused to the transmembrane and intracellular domain of XA21 to form a chimeric receptor that responds to elf18 treatment and confers partial resistance to Xoo in transgenic rice lines (Schwessinger et al., 2015a).

The availability of rapid and reliable assays that measure markers characteristic of immune response activation can help facilitate investigations of innate immune signaling. For example, immune signaling studies of FLS2 and EFR in Arabidopsis have been aided by the availability of rapid and reliable assays (Gomez-Gomez & Boller, 2000; Zipfel et al., 2006; Chinchilla et al., 2007; Lu et al., 2010; Albert et al., 2010; Schulze et al., 2010; Schwessinger et al., 2011; Sun et al., 2013; Li et al., 2014). In contrast, studies of the XA21-mediated immune response have been limited by the lack of rapid assays and well-characterized genetic markers. Typically, disease assessments are carried out by measuring lesions on rice leaves or by assessing bacterial populations from infected leaves (Kauffman et al., 1973; Song et al., 1995; Da Silva et al., 2004; Park et al., 2008; Chen et al., 2014; Pruitt et al., 2015).

In this study we aimed to establish a rapid and efficient assay to monitor the XA21-mediated immune response after bacterial infection. For this purpose, we employed the EFR:XA21:GFP chimera composed of the EFR extracellular domain and the XA21 transmembrane and intracellular kinase domains, tagged with green fluorescent protein (EFR:XA21:GFP) (Schwessinger et al., 2015a). EFR:XA21:GFP transgenic rice plants are partially resistant to Xoo and detached EFR:XA21:GFP leaves respond to elf18 with stress-related gene induction, mitogen-activated protein kinase (MAPK) cascade activation, and reactive oxygen species (ROS) production (Schwessinger et al., 2015a). These results indicate that plants expressing the EFR:XA21:GFP chimeric protein are appropriate for studies to identify markers of resistance.

We used RNA sequencing (RNAseq) to identify genes differentially regulated in elf18 treated EFR:XA21:GFP rice. We then assessed if differentially regulated genes (DRGs) in elf18 treated EFR:XA21:GFP rice leaves were up-regulated in Xoo infected rice leaves expressing full-length XA21, which are resistant to Xoo. We developed a rapid and reliable assay to analyze gene expression in detached rice leaves inoculated with Xoo. We identified 8 DRGs from elf18 treated EFR:XA21:GFP rice that are also specifically up-regulated in detached XA21 rice leaves infected with Xoo.

Materials and Methods

Plant growth, peptide and bacterial treatments of detached rice leaves

For peptide treatments, wild type (WT) Kitaake and progeny from line EFR:XA21:GFP-3-8 (EFR residues 1-649 and XA21 residues 651-1025 called EFR:XA21:GFP in this study) Kitaake rice leaves were harvested from plants grown in the greenhouse for 4.5 weeks (Schwessinger et al., 2015a). 1.5–2 cm leaf sections were collected from expanded adult leaves using surgical grade scissors. Tissue from the leaf base and leaf tip was discarded. Detached leaves were equilibrated overnight in 6-well Costar cell culture plates under constitutive light (between 5–10 µmol/(m2*s)) (Fig. S1). Peptide treatments were performed in the morning and collected at the indicated times.

For bacterial inoculations, we used detached rice leaves harvested from 4-week old plants grown using a hydroponic growth system as described previously (Pruitt et al., 2015) under a light intensity of 280 µmol/(m2*s). Freshly harvested leaves from Kitaake and Ubi-Myc:XA21 Kitaake rice (called Myc:XA21 rice in this study) (Park et al., 2010) were cut into 1.5–2 cm pieces and immediately (without overnight equilibration) floated on 10 mM MgCl2 solution for mock treatments or 10 mM MgCl2 containing fresh Xoo cell suspensions at O.D.600 of 0.1 (approximately 1 × 107 cells mL−1). The samples were left overnight under constitutive light (between 5–10 µmol/(m2*s)) and collected 24 h post-inoculation (hpi). Leaves were floated on approximately 1.5 mL Xoo cell suspension media in 6-well Corning Costar cell culture plates (Fig. S1). The detached leaf infection assay allows a more uniform distribution, compared to the scissor inoculation method (Kauffman et al., 1973), of Xoo inoculum by floating leaves on bacterial suspensions.

RNA isolation and qPCR gene expression analysis for peptide treated and bacterial infected leaf samples

Detached leaves were frozen in liquid nitrogen and powdered using a Qiagen tissuelyser. For tissue from greenhouse grown plants, RNA was extracted from powdered tissue using TRI Reagent and precipitated with isopropanol. For tissue from hydroponically grown plants, RNA was extracted using the Spectrum Plant Total RNA Kit from Sigma-Aldrich. RNA was DNase treated using the TURBO DNase kit from Life Technologies. RNA concentrations were normalized to the lowest sample concentration in each experiment. cDNA was synthesized from 2 µg of total RNA using the High Capacity cDNA Reverse Transcription Kit by Life Technologies. Gene expression changes were determined by ΔΔCt method (Schmittgen & Livak, 2008) normalized to Actin (LOC_Os03g50885) and compared to mock treated samples.

Identification of genes differentially regulated in elf18 treated EFR:XA21:GFP rice using RNA sequencing

Plant growth, leaf tissue isolation, and treatments were performed as described above. RNA was isolated from untreated Kitaake as well as untreated and elf18 treated EFR:XA21:GFP leaf tissue using the Spectrum Plant Total RNA Kit from Sigma-Aldrich and on-column DNase treated to remove genomic DNA contamination following the manufacturer’s instructions. RNA was quantified using the Quant-IT Ribogreen RNA Assay Kit. Sequences were deposited to the NCBI Sequence Read Archive (BioProject ID PRJNA250865).

RNAseq libraries, sequencing, and reference alignment were performed as described previously (Schwessinger et al., 2015a). Sample correlation between Kitaake and EFR:XA21:GFP replicates and differential gene expression analysis was performed using the Bioconductor ‘edgeR’ package for R (Robinson, McCarthy & Smyth, 2010; McCarthy, Chen & Smyth, 2012).

Bacterial strains and generation of mutants

We generated a PXO99AΔhrpA1 mutant in Philippine race 6 strain PXO99Az, a derivative of strain PXO99 (referred to as PXO99A in this study) (Salzberg et al., 2008). Xoo was grown in 10 g PSB (10 g Peptone (bacto-Peptone), 10 g Sucrose, 1 g sodium glutamate (glutamic acid, monosodium salt), final volume 1L, pH 7.0) or on PSA plates (PSB with 16 g/L bacto-agar) at 28°C. PXO99AΔhrpA1 was generated by single crossover mutagenesis using the suicide vector pJP5603 (Penfold & Pemberton, 1992). An approximately 500 base pair sequences within hrpA1 was amplified using forward primer 5′-CGGGGTACCGTGCTGCGTGATTTGTCCG-3′and reverse primer 5′-CGCGGATCCTGACTTGGTCGATGCAGTCC-3′and cloned into the multiple cloning site of pJP5603 using the restriction enzyme sites KpnI and BamHI. PXO99A-competent cells were transformed with the suicide plasmids by electroporation and plated to PSA with kanamycin (50 µg/ml). PXO99AΔhrpA1 colonies with kanamycin resistance were screened by PCR for colonies with single crossover events, which contain the vector disrupting the target gene. PXO99AΔraxST and PXO99AΔraxST(raxST) complemented strains used in this study were described previously (Pruitt et al., 2015). PXO99AΔraxST evades XA21-mediated immunity while the complemented PXO99AΔraxST(raxST) strain does not.

Results

RNAseq analysis identifies 2212 genes that are differentially regulated in elf18 treated EFR:XA21:GFP rice leaves

We analyzed the transcriptomic profile of EFR:XA21:GFP rice lines treated with elf18 to identify genes differentially regulated during this response. We sequenced cDNA from EFR:XA21:GFP leaves treated with 500 nM elf18 for 0.5, 1, 3, 6, and 12 h. We also included untreated EFR:XA21:GFP and Kitaake as controls (Table S1). Multidimensional scaling of pairwise biological coefficient of variance comparisons for each sample revealed that replicate samples group together (Fig. 1A). This grouping of biological replicates demonstrates the overall transcriptional similarity between each sample (Robinson, McCarthy & Smyth, 2010).

Figure 1 The transcriptomic profile of elf18 treated EFR:XA21:GFP rice is enriched for stress response related and photosynthesis-related genes.

(A) Multi-dimensional scaling comparing biological coefficients of variance between each sample. Samples labeled Kit0 are Kitaake rice leaf samples without treatment. Samples labeled Kitaake represent untreated Kitaake samples at 0 h, EFRX represent EFR:XA21:GFP untreated samples (EFRX0) and samples treated with 500 nM elf18 at 0.5 h (EFRX0.5), 1 h (EFRX1), 3 h (EFRX3), 6 h (EFRX6), and 12 h (EFRX12). Groups of technical replicates are circled and sample color codes are indicated in upper left legend. (B) A five-way Venn diagram indicating number of total (indicated in parentheses), unique and overlapping differentially regulated genes between time points. (C) Heatmap representing expression levels of differentially regulated genes (DRGs) for EFR:XA21:GFP samples treated with elf18 for indicated durations. The three major DRG clades, determined by expression profile, are labeled 1, 2 and 3 and are indicated to the right of the heatmap. Significantly enriched gene ontology terms with a false discovery rate less than 0.5, compared to the reference, for each clade are shown on the right under the respective clade number. The heatmap color key indicates log2 fold change values compared with untreated, EFR:XA21:GFP 0 h samples.

We identified 2,212 genes that were differentially regulated in EFR:XA21:GFP rice treated with elf18 compared with untreated (0 h) samples. Using a false discovery rate (FDR) cutoff of 0.05 and absolute expression log fold change (logFC) of 2 or greater, we previously reported that the transcriptomic profile of untreated Kitaake compared to untreated EFR:XA21:GFP did not differ significantly (Schwessinger et al., 2015a). Over the treatment time course, we identified 2,212 DRGs (FDR <0.05, absolute logFC >2) using untreated EFR:XA21:GFP at 0 h as a reference. The number of DRGs that overlap between the elf18 treatment time points are summarized in Fig. 1B and File S1. Of the 2,212 differentially regulated genes, there were 1,420 up-regulated and 792 down-regulated genes. The highest number of DRGs (1,504) was observed 6 h post elf18 treatment. These results show that elf18 treated EFR:XA21:GFP rice express a substantially different set of genes over time compared to untreated (0 h) samples.

Stress response related genes are up-regulated while photosynthesis related genes are down-regulated in EFR:XA21:GFP rice treated with elf18

To examine the types of biological processes affected in elf18 treated EFR:XA21:GFP rice, we analyzed GO term enrichment of DRGs using the AgriGo analysis tool (Du et al., 2010). A total of 1,204 out of 1,420 of the up-regulated DRGs and 682 of the 806 down-regulated DRGs had GO annotations. An FDR of 0.05 or less was used to define significantly enriched terms compared to the Michigan State University annotation reference as calculated by the AgriGo tool (Du et al., 2010; Kawahara et al., 2013). Fig. 1C and File S2 summarize the most enriched GO terms in each of the three major DRG clades. Clade 1 contains 1,333 genes that are mostly up-regulated over time. Genes from clade 1 are enriched for metabolic process (GO:0008152), response to stimulus (GO:0050896) and response to stress (GO:0006950) GO terms (Fig. 1C). Clade 2 genes (122) are up-regulated across all time points and are enriched for secondary metabolic process (GO:0019748), metabolic process (GO:0008152) and response to stress (GO:0006950) GO terms (Fig. 1C). Clade 3 consists of 757 genes that are mostly down-regulated in all timepoints. Photosynthesis (GO:0015979) and response to abiotic stimulus (GO:0009628) are the most enriched GO terms associated with clade 3 genes (Fig. 1C).

qPCR validation of genes up-regulated in elf18 treated EFR:XA21:GFP plants

We chose 23 DRGs from the elf18 treated EFR:XA21:GFP rice RNAseq dataset with relatively high logFC and low FDR values after 3, 6, and 12 h for detailed analysis. We first assessed if the 23 genes up-regulated in elf18 treated EFR:XA21:GFP could be validated by qPCR analysis. Eleven out of 23 DRGs were up-regulated in EFR:XA21:GFP rice leaves after elf18 treatment. Transcripts of the remaining 12 candidate genes were detectable by qPCR amplification but were not up-regulated in elf18 treated EFR:XA21:GFP leaves (File S3).

Establishment of bacterial infection assay of detached rice leaves

We established a detached leaf infection assay to test if genes identified in the EFR:XA21:GFP experiments are representative of genes differentially regulated in Xoo infected Myc:XA21 rice. We observed bacterial ooze from the detached rice leaves three days after inoculation with Xoo strain PXO99A (Fig. 2). To further assess if Xoo infects rice leaves in our detached leaf infection assay, we measured the expression level of Os8N3 (LOC_Os08g42350), which was previously shown to be up-regulated in rice upon Xoo infection and is thus a useful marker of successful infection (Yang, Sugio & White, 2006). For these experiments, we also included a mutant PXO99A strain (PXO99AΔhrpA1) that is unable to infect rice as a control. The hrpA1 gene encodes a pilus protein that is essential for type III-secretion of effectors required for host infection (Wengelnik et al., 1996). We observed that the PXO99AΔhrpA1 Xoo mutant is unable to infect Kitaake and Myc:XA21 rice plants (Fig. S2). Both WT Kitaake and Myc:XA21 detached leaves express Os8N3 at higher levels compared to mock treatments 24 hpi with WT PXO99A, but not with PXO99AΔhrpA1 (Fig. 3). These results demonstrate that Xoo infects detached rice leaves.

Figure 2 Bacterial oozes from an infected rice leaf.

Bacterial oozing (white arrowheads) was observed from rice leaf xylem vessels three days post infection. This image shows detached Kitaake rice leaves infected with PXO99A in a 6-well cell culture plate. Bacterial oozing was consistently observed in Kitaake and Myc:XA21 detached leaves infected with PXO99A. Rice leaves were collected from 4-week old, hydroponically grown plants and floated on Xoo cell suspension media.

Figure 3 A marker gene of Xanthomonas infection, Os8N3, is up-regulated in PXO99A infected leaves.

Os8N3 expression in detached Kitaake and Myc:XA21 rice leaves with 10 mM MgCl2 mock treatment or infected with PXO99A or PXO99AΔhrpA1 (ΔhrpA1) at an O.D.600 of 0.1. Letters represent statistically significant differences between mean expression values (p < 0.05) determined by using a Tukey–Kramer HSD test. This experiment was repeated three times with similar results.

We next employed the detached leaf infection assay to examine the expression of the stress-related marker PR10b in Xoo infected Myc:XA21 rice leaves. Compared with mock treated controls, PR10b is up-regulated in flg22 treated rice, elf18 treated EFR:XA21:GFP rice and Myc:XA21 rice treated with the RaxX21-sY (Chen et al., 2014; Schwessinger et al., 2015a; Pruitt et al., 2015). Using qPCR, we detected significant up-regulation of PR10b expression in Myc:XA21 rice leaves 24 hpi with PXO99A and PXO99AΔhrpA1. PR10b up-regulation was not observed in infected Kitaake leaves (Fig. 4). These results show that the detached leaf infection assay can be used to assess XA21-mediated marker gene expression and also indicate that RaxX expression or secretion is not affected by the ΔhrpA1 mutation.

Figure 4 The stress-related marker gene PR10b is up-regulated in Xanthomonas infected XA21 rice.

PR10b expression in detached Kitaake and Myc:XA21 rice leaves with 10 mM MgCl2 mock treatment or infected with PXO99A or PXO99AΔhrpA1 (ΔhrpA1) at an O.D.600 of 0.1. Letters represent statistically significant differences between mean expression values (p < 0.05) determined by using a Tukey–Kramer HSD test. This experiment was repeated three times with similar results.

Figure 5 Eight marker genes are specifically up-regulated in detached rice leaves undergoing the XA21-mediated immune response.

Expression of eight genes was measured in detached Kitaake and Myc:XA21 leaves infected with different Xoo strains. Mock samples were treated with 10 mM MgCl2. Xoo strains used for infection were WT PXO99A, a PXO99AΔraxST mutant strain that evades XA21-mediated immunity (ΔraxST), and the PXO99AΔraxST mutant strain complemented with raxST (ΔraxST (raxST)). Expression levels are normalized to Actin then compared to mock treated samples. Bars indicate mean expression levels ± standard deviation of three technical replicates. Letters represent statistically significant differences between mean expression values (p < 0.05) determined using a Tukey–Kramer HSD test. This experiment was repeated twice with similar results.

Eight out of 11 genes induced in efl18 treated EFR:XA21:GFP rice are up-regulated in Xoo infected XA21 rice

We next employed the detached leaf infection assay to identify genes up-regulated upon Xoo infection of Myc:XA21 rice leaves. For these assays, we examined the gene expression of the 11 genes validated by qPCR analysis of elf18 treated EFR:XA21:GFP rice described above. We inoculated Kitaake and Myc:XA21 rice with WT PXO99A, PXO99AΔraxST mutants, and PXO99AΔraxST complemented with raxST (PXO99AΔraxST(raxST)). Xoo strains carrying mutations in raxST do not activate XA21-mediated immunity (Da Silva et al., 2004; Pruitt et al., 2015). The expression of 8 of 11 genes was specifically up-regulated in detached Myc:XA21 rice leaves 24 hpi with PXO99A and PXO99AΔraxST (raxST) but not in Myc:XA21 rice leaves infected with PXO99AΔraxST (Fig. 5 and File S3). The 8 validated marker genes encode a putative subtilisin-like protein (LOC_Os04g03100), a reticuline oxidase-like protein precursor (LOC_Os06g35700), the decarboxylase OsTDC1 (LOC_Os08g04540) (Kang et al., 2007), a peroxidase precursor (LOC_Os11g02100), RsOsPR10a (LOC_Os12g36830) (Hashimoto et al., 2004; Takeuchi et al., 2011), CYP71Z7 (LOC_Os02g36190) (Li et al., 2013), OsKO5 (LOC_Os06g37224) (Itoh et al., 2004), and one protein without a putative function (LOC_Os10g28299). The 3 remaining genes that were up-regulated in elf18 EFR:XA21:GFP rice but not in Myc:XA21 rice leaves encode a isoflavone reductase (LOC_Os01g13610), a subtilisin-like protein (LOC_Os04g03100), and a reticuline oxidase-like protein precursor (LOC_Os06g35700).

Discussion

In this study we identified 8 genes that are specifically up-regulated in both elf18 treated EFR:XA21:GFP and Xoo infected detached Myc:XA21 rice leaves. At the time of these experiments, the activator of XA21, RaxX, had not yet been identified (Pruitt et al., 2015). We therefore treated rice plants expressing the EFR:XA21:GFP chimera with elf18 to identify candidate marker genes because EFR:XA21:GFP rice are partially resistant to Xoo and respond to elf18 treatments as described above in the introduction. Our results show that even though the EFR:XA21:GFP-mediated response does not confer robust resistance to Xoo (Schwessinger et al., 2015a), similar genes are up-regulated during both EFR:XA21:GFP- and Myc:XA21-mediated responses (Fig. 5). Further studies are necessary to determine why the expression of EFR:XA21:GFP in rice does not confer robust resistance to Xoo.

We show that stress-related gene induction of PR10b in Myc:XA21 rice leaves is maintained in plants inoculated with PXO99AΔhrpA1 mutant strains. These results suggest that RaxX expression, modification and secretion is not compromised by the ΔhrpA1 mutation. These results indicate that RaxX function is independent of type-III secretion mediated by hrpA1. It was previously reported that the raxSTAB operon, which encoded predicted components of a type-I secretion system, was required for the processing and secretion of the XA21 elicitor (Da Silva et al., 2004). Our finding that RaxX function is independent of hrpA1-mediated type-III secretion is consistent with the hypothesis that RaxX is a type I-secreted molecule (Da Silva et al., 2004; Pruitt et al., 2015) and may provide insight into the largely unknown biological function of RaxX.

The discovery of RaxX and the establishment of the detached leaf infection assay described here provide useful tools for studying XA21-mediated immunity. XA21 activation can be measured through ROS production and marker gene expression in detached leaves treated with the RaxX21-sY peptide (Pruitt et al., 2015; Schwessinger et al., 2015b). One advantage of this approach is that researchers can study XA21-mediated immunity without working with Xoo. Instead, researchers can activate XA21-mediated immunity by treating leaves with RaxX21-sY peptide rather than Xoo. This strategy eliminates the need for select agent permits, which are costly and time-consuming. The assay described in this study now allows researchers to use Xoo infected plants to monitor XA21 activation by gene expression, which was previously only possible using peptide treatments. This provides the benefit of monitoring bacterial induced genes, such as Os8N3 (Fig. 3). While we are not able to definitively assess resistance versus susceptibility to Xoo using this assay, we demonstrate that we can use gene expression to monitor an immune response specifically mediated by XA21.

The detached leaf infection assay can also be used for other studies of bacterial-rice interactions. For example, this system can be used to study rice immune responses conferred by different resistance genes or induced by different bacterial strains. For example, the detached leaf infection assay can be used to study the immune response conferred by other rice Xa genes (Khan, Naeem & Iqbal, 2014) that confer resistance to Xoo such as Xa3/Xa26, which also encodes a cell surface receptor kinase (Xiang et al., 2006; Li et al., 2012). The detached leaf infection assay can also be adapted to study immune responses to other races of Xoo (Niño-Liu, Ronald & Bogdanove, 2006) or other Xanthomonas pathovars such as Xanthomonas oryzae pv. oryzicola (Raymundo, Perez & Co, 1992; Niño-Liu, Ronald & Bogdanove, 2006).

Supplemental Information

Figure S1 Image of Xoo infection of detached rice leaves

Image of detached rice leaf assay setup. 1.5-2cm detached leaves are floated on 1.5mL of bacterial suspension in 6-well flat bottom cell culture plates (approximately 12.5 × 8.5 × 2 cm).

Click here for additional data file.

Figure S2 Infection with PXO99AΔhrpA1 mutants does not form lesions on Kitaake or XA21 rice leaves

Kitaake or Myc:XA21 rice were inoculated with scissors dipped in PXO99A or PXO99AΔhrpA1 (ΔhrpA1) at an approximate cell density of 8x108 cells mL-1. Boxplots (red) represent distribution of lesion measurements from three different plants taken 14 days after infection with at least three measurements from each plant (n ≥ 9). Blue lines indicate standard deviation of the mean.

Click here for additional data file.

Table S1 RNA sequencing sample treatment summary

Table summarizes the experimental setup including the genotypes, time of treatment, and type of treatment used for samples used in RNA sequencing. There were three replicates for each sample for a total of 21 sequenced samples.

Click here for additional data file.

File S2 Differentially regulated gene clade groups and GO term analysis

Click here for additional data file.

File S1 Differentially regulated gene expression table with clade number and putatuve function

Click here for additional data file.

File S3 Marker gene summary and primer information

Click here for additional data file.

We would like to thank Vasanth R. Singan, Rita Kuo, Mansi Chovatia, and Christopher Daum from the Joint Genome Institute for their help with RNA sequencing. The United States Government retains and the publisher, by accepting the article for publication, acknowledges that the United States Government retains a non-exclusive, paid-up, irrevocable, world-wide license to publish or reproduce the published form of this manuscript, or allow others to do so, for United States Government purposes.

Additional Information and Declarations

Competing Interests

Author Contributions

Data Availability

Pamela C. Ronald is an Academic Editor for PeerJ.

Nicholas C. Thomas performed the experiments, analyzed the data, wrote the paper, prepared figures and/or tables, reviewed drafts of the paper.

Benjamin Schwessinger conceived and designed the experiments, performed the experiments, analyzed the data, wrote the paper, reviewed drafts of the paper.

Furong Liu, Huamin Chen, Yen P. Nguyen and Isaac W.F. Shaker performed the experiments, reviewed drafts of the paper.

Tong Wei analyzed the data, reviewed drafts of the paper.

Pamela C. Ronald conceived and designed the experiments, analyzed the data, contributed reagents/materials/analysis tools, reviewed drafts of the paper.

The following information was supplied regarding data availability:

The National Center for Biotechnology Information Sequence Read Archive (SRA)

BioProject ID PRJNA250865.

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
