# Peer review of "XA21-specific induction of stress-related genes following Xanthomonas infection of detached rice leaves"

_PeerJ, doi:10.7717/peerj.2446_

## Round 0.1 · original submission · Minor Revisions

Please incorporate/address the suggestions from the reviewers, or explain to me if you think any cannot be addressed.

Reviewer 1 ·

Basic reporting

Generally this was well done.

A few more sentences about the role of type I secretion system in delivering RaxX might strengthen the significance of this finding (lines 240-245).

line 234: word missing, perhaps “EFR:XA21:GFP lines”
line 241: “in plants inoculated with”
line 248-249: I did not see any data supporting that ROS production is induced after treatment with the RaxX21-sY peptide in the detached leaf assay. This should be rephrased as a future aim, or the data should be shown to demonstrate that it works.
line 447: Figure 5 legend is incomplete.

Experimental design

Pruitt et al recently showed that RaxX is required to induce Xa21 immunity. As RaxX is located in a different operon from RaxSTAB, it is possible that it contributes differently to marker gene expression (Figure 5). It would be good to know that the marker genes respond in the same way to RaxX as they do to RaxST, particularly if this will be used as a surrogate system to infection, as discussed on lines 246-253.

Validity of the findings

The detached leaf assay could potentially be a powerful system to investigate the molecular mechanisms. However it was not completely convincing to me that the detached leaf assay can distinguish between resistance and susceptibility. For example, the Figure 2 legend indicates that bacterial ooze was observed from both Kitaake and Myc:Xa21 lines infected with PX099A, and Figure 3 shows similar levels of Os8N3 gene induction in either genotype in response to PX099A. Since Kitaake is supposed to be susceptible and Myc:Xa21 is supposed to be resistant, it seems that a difference should be observed in both experiments. For Figure 2, quantitation of the level of bacterial growth in the two genotypes and a difference between the two genotypes would more convincingly demonstrate resistance versus susceptibility. In addition, the authors should include additional negative controls in Figure 3, i.e. rice cultivars that are known not to respond to Os8N3, to ensure marker gene expression is unaffected by the environment.

Additional comments

The authors previously showed that transgenic plants carrying a gene encoding the EFR LRR domain, Xa21 transmembrane and intracellular domain and GFP (EFR:Xa21:GFP) respond to the elf18 PAMP and provide partial immunity to Xanthomonas oryzae pv. oryzae (Xoo). In this manuscript, using the EFR:Xa21:GFP reporter lines, they identified candidate genes that are activated by Xa21 using RNAseq. They chose 23 differentially regulated genes (DRGs) for further analysis and validated 11/23 by qPCR. The authors developed a detached leaf assay to evaluate induction of the Xa21-mediated immune response in rice. Using the detached leaf assay, they showed that PR10b gene expression was induced in resistant Myc:Xa21 lines infected with PX099A or a ∆hrpA1 mutant. 8/11 DRGs were induced in Myc:Xa21 lines treated with PX099A or PX099A∆raxST(raxST) but not PX099A∆raxST or Kitaake lines with any of the treatments, suggesting that these genes are specifically induced during resistance in response to RaxST. The other 3 DRGs appear to have been specific to EFR:Xa21:GFP instead of Xa21.

Reviewer 2 ·

Basic reporting

I do not see reference to the raw data RNAseq experiment. These files can be deposited to the NCBI Sequence Read Archive or to JGI's portal as in Schwessinger et al 2015a

Experimental design

No comments. Experimental designs are well explained and the paper has a thorough materials and methods section.

Validity of the findings

No comments. New data on elicitation of an XA21 chimera are well presented and primers/locus numbers for helpful elicitation markers are given.

Additional comments

The manuscript is clearly written and presents several well-designed experiment on rice plants elicited via the XA21 pathway. It also details a protocol for successful infection of detached rice leaves with Xanthomonas oryzae. Genes upregulated over a timecourse of elicitation in transgenic rice expressing an elf18 triggered EFR::XA21 chimera are identified by RNAseq. Marker candidates are validated by qPCR and confirmed to be regulated similarly in a bacterial resistance reaction using XA21 transgenic rice. Controls are abundant and appropriate for the detached leaf experiment, relying on the group’s recent rax21 findings. The figures are clear and primers for these XA21 markers are helpfully presented.

Specific comments:
Line 73: It would be helpful here to restate the amino acid cutoffs for the chimeric break.
Line 187: Do lesions eventually form in the detached leaf assay?
Line 248: this paragraph states the “infection assay” is novel and useful, but goes on to describe advantages of peptide treatment of detached leaves, not bacterial infection. The peptide treatment has been performed by this group previously. Are there research advantages to involving the bacteria?
Fig 1: Could the sum of DRGs for each timepoint be presented, maybe on the outside of the Venn diagram?

---

## Round 0.2 · accepted · Accept

Thank you for addressing the reviewer comments.